# Similar Cognitive Skill Impairment in Children with Upper Limb Motor Disorders Due to Arthrogryposis Multiplex Congenita and Obstetrical Brachial Plexus Palsy

**DOI:** 10.3390/ijerph20031841

**Published:** 2023-01-19

**Authors:** Evgeny Blagovechtchenski, Maria Koriakina, Dimitri Bredikhin, Olga Agranovich, Dzerassa Kadieva, Evgenia Ermolovich, Iiro P. Jääskeläinen, Anna N. Shestakova

**Affiliations:** 1Centre for Cognition & Decision Making, Institute for Cognitive Neurosceince, National Research University Higher School of Economics, 101000 Moscow, Russia; 2Federal State Budgetary Institution, The Turner Scientific Research Institute for Children’s Orthopedics under the Ministry of Health of the Russian Federation, 196603 St. Petersburg, Russia; 3Brain and Mind Laboratory, Department of Neuroscience and Biomedical Engineering, Aalto University School of Science, 00076 Espoo, Finland

**Keywords:** cognitive function, arthrogryposis, obstetrics palsy, children, motor disorder

## Abstract

Arthrogryposis multiplex congenita (AMC) and obstetrical brachial plexus palsy (OBPP) are motor disorders with similar symptoms (contractures and the disturbance of upper limb function). Both conditions present as flaccid paresis but differ from each other in the pathogenesis: AMC is a congenital condition, while OBPP results from trauma during childbirth. Despite this difference, these diseases are identical in terms of their manifestations and treatment programmes. We compared the cognitive skills of children with AMC and OBPP diagnoses with those of healthy children; we also compared the motor skills of impaired children with those of healthy ones. The patients in both groups significantly differed from the healthy children with regard to psychological parameters, such as ‘visual memory capacity’ and ‘thinking’. Moreover, the two groups with children with AMC and OBPP significantly differed from each other in motor skill parameters, such as ‘delayed motor development’, ‘general motor development’, and the ‘level of paresis’. Upper limb motor function in the OBPP children was less impaired compared to that of the AMC children. However, we did not find any significant differences in cognitive deficits between the AMC children and the OBPP children. This may indicate that motor impairment is more significant than the underlying cause for the development of cognitive impairment; however, the factors causing this phenomenon require further study (e.g., social environment, treatment, and rehabilitation programme).

## 1. Introduction

Motor skills and cognitive skills are often discussed together in the context of human development. However, the extent to which one affects the other remains unclear. For example, the question of whether it is worth changing the educational programme for children with motor disabilities has not yet been resolved. In this study, we attempted to find a clearer answer to the following question: what is more important for the child—the diagnosis itself or its manifestation?

Motor development plays a critical role in children’s understanding of the physical and social worlds [1]. However, the extent to which motor development affects cognitive skills remains unclear [2,3,4]. For example, the association between motor planning and working memory performance was shown [5]. In school-age children, a correlation has been found between physical activity and cognitive skills [6]. It can be hypothesised that special school programmes must be developed for children with motor deficits. However, in most countries, educational programmes for children with motor disorders are the same as those for healthy children.

In this study, we focused on diseases associated with impaired functioning of the upper extremities. The upper limbs in humans have a special functional role: unlike the lower limbs, they are mainly involved in the performance of precise voluntary movements. In performing voluntary movements, all levels of the nervous system are involved, especially the highest—the cerebral cortex [7,8,9]. Cognitive skills are also associated with the involvement of higher levels of control in the brain. This may be the critical factor that determines the interaction between the level of development of cognitive skills and motor skills (associated with precise voluntary movements). Cognitive skills and motor skills exhibit a similar developmental timeline, especially between the ages of 5 and 10 years; they also share several important psychophysiological processes, such as sequencing, monitoring, and planning [10,11].

In this study, we assessed the cognitive skills and motor skills of children with upper limb movement disorders, specifically, congenital arthrogryposis multiplex (AMC) and obstetric brachial plexus palsy (OBPP). AMC refers to a group of congenital conditions characterised by joint contractures in two or more areas of the body. While the precise cause may be unknown for some individuals, the causes of AMC are variable and may include genetic, parental, and environmental factors as well as abnormalities that form during foetal development [12]. Individuals with AMC have limited joint movement, with or without muscle weakness, in the affected areas of the body. Contractures vary in distribution and severity and do not progress to previously unaffected joints, but they may change over time due to growth and treatment. The lack of active movement in the joints of the upper extremities is one of the main problems that limit or prevent self-care [13]. AMC is considered a congenital condition, and its pathology is not sufficiently clear. OBPP is an injury to the brachial plexus that occurs during childbirth, usually as a result of strain during a difficult vaginal delivery. This leads to the paralysis of the upper limb; thus, this condition is not congenital, unlike AMC [14]. A factor that was important to this study is that both diseases manifest in the same manner—similar dysfunctions of the upper limbs. Furthermore, surgical interventions and the therapy processes following such interventions are similar. In clinical practice, motor skills are restored through the autotransplantation of muscles from various donor areas [15]. Postoperative therapy involves therapeutic physiotherapy and special physical exercises. Notably, there is no difference between the recovery processes for either of these diseases [14]. Some cognitive skills in children have been shown to be associated with motor illnesses. Compared to healthy children, the capacity for visual and auditory memory is lower in patients with AMC or OBPP. This is particularly evident in children aged 8–11 years [16]. However, the extent to which the degree of movement impairment affects cognitive skills remains unclear.

In this study, we compared the cognitive skills of children with similar manifestations of upper limb motor dysfunction (including treatment and therapy methods) with the cognitive skills of healthy children of a similar age. The patients with OBPP initially had a higher level of motor skill development. We also compared how this difference in motor skills is reflected in the degree of development of cognitive skills. We also correlated the severity of the motor impairment with the level of cognitive performance.

## 2. Methods

### 2.1. Patients and Control Participants

This study had 28 control participants (16 males, 12 females; mean age ± std: 9.95 ± 3.38), 18 amyoplasia participants (10 males, 8 females; mean age ± std: 10.14 ± 2.40), and 11 OBPP participants (6 males, 5 females; mean age ± std: 10.64 ± 2.54).

### 2.2. Assessment of Cognitive Functions and Motor Functions

Cognitive functions (attention span, auditory memory, visual memory, conceptual development, and thinking) were measured using a battery of diagnostic techniques.

Attention and auditory memory were assessed using the Wechsler Intelligence Scale for Children (WISC). The WISC-IV was used for children over six years of age, and the Wechsler Preschool and Primary Scale of Intelligence (WPPSI) was used for children aged 3–6 years [17]. Attention was reflected in a child’s ability to repeat backward the numbers they had heard, and auditory working memory was reflected in the number of digits the child had memorised.

Intelligence was evaluated using Raven’s progressive matrices (A, B, and C) [18]. We employed two types of tests, namely the CPM/CVS kit and SPM+/MHV, because of the age range of the children in the study. Intelligence was reflected in the number of correct responses to age-appropriate intellectual ability tasks.

Visual memory and conceptual development as an aspect of thinking were measured using Shipitsina’s ‘psychological diagnostics of deviations in the development of children of primary school age’. Visual working memory was reflected in the number of memorised pictures out of the 10 presented. Conceptual development was assessed by the number of points a child scored when composing a story from pictures.

Thinking was assessed using the ‘exclusion of objects’ technique. This technique is designed to study the features of thinking—first, the level of development and second, the qualitative characteristics of the generalisation process of visual materials [19].

During neurology examinations, the patient’s general motor skills (GMD) were estimated. Anamnesis vitae included information about the patient in infancy. A developmental delay was indicated when a child had not reached particular milestones within the expected time period and his motor skills were different from those of a healthy child (DMD). We estimated passive and active movement in the joints, muscles strength, muscle volume, muscle tone, tendon reflexes, and sensation. The paresis level was estimated clinically by the scheme of segmental innervation of upper limb muscles. A lower level of paresis is associated with greater motor impairment in the patient, i.e., greater involvement of the distal muscles [15].

### 2.3. Statistical Analysis

The Kruskal–Wallis H test was used to test the hypothesis that population median performances in different cognitive tasks are equal among AMC patients, OBPP patients, and control children. Mann–Whitney tests were used to investigate the exact differences between AMC patients, OBPP patients, and control children.

## 3. Results

The patients were examined during their spare time. They attended the laboratory supervised either by their parents or medical personnel. Their self-care capabilities were significantly different from healthy children. The patients could not take food or perform hygiene actions (wash face and hands) on their own. Rather, they did so under the patronage of their parents or medical personnel. We also noticed that all the movements performed by the patients appeared to be slower compared to healthy children. We plan to quantify these skills. Visual communication did not differ between healthy children and patients.

Comparison of children with AMC, OBPP and healthy children based on the Kruskal–Wallis H-test showed that significant deviations were observed for such parameters as «Visual Memory» and «Thinking» (Table 1).

**Table 1 ijerph-20-01841-t001:** Results of Kruskal–Wallis H-test series testing the hypothesis that population median performances in different cognitive tasks are equal between AMC patients, OBPP patients, and control children. *p*-values are adjusted by means of Bonferroni correction (included eta-squared values for the Kruskall–Wallis series with H-values following Murphy and Myors ([20], Appendix A), ‘*’—statistically significant values).

Task	h	η^2^_approximated_	*p*
Attention	2.38	0.04	1.000
Auditory Memory	1.83	0.03	1.000
Visual Memory	9.90	0.16	0.042 *
Intelligence	1.62	0.03	1.000
Conceptual Development	2.65	0.05	1.000
Thinking	12.76	0.19	0.010 *

Mann–Whitney tests were used as post hoc analyses to investigate the exact differences between AMC patients, OBPP patients, and control children. Specifically, with regard to the visual memory performance, both AMC (u = 369.0, *p* = 0.023) and OBPP (u = 230.5, *p* = 0.046) differed significantly from their control peers, whereas AMC and OBPP patients showed equal results (u = 102.5, *p* = 1.00). Similarly, with regard to the thinking task performance, both AMC (u = 353.5, *p* = 0.012) and OBPP (u = 237.0, *p* = 0.004) differed significantly from their control peers, whereas AMC and OBPP patients showed equal results (u = 109.0, *p* = 1.00) (Figure 1).

We also assessed the difference in motor skills between children with AMC and OBPP diagnoses (Figure 2). The series of Mann–Whitney tests revealed that OBPP children showed significantly higher motor performance than AMC children. Specifically, they demonstrated higher DMD scores (u = 24.4, *p* = 0.001), higher GMD scores (u = 10.5, *p* < 0.001), and higher level of paresis (u = 21.0, *p* < 0.001).

## 4. Discussion

In this study, we evaluated the differences in cognitive skills between children with motor disorders of the upper limbs and healthy children; we then further correlated these factors. This study involved three groups of children (adjusted for age): patients with AMC, patients with OBPP, and healthy children. Despite the similarity of disorders between the patients, the statistical analysis showed a significant difference in the level of motor skills between children diagnosed with AMC and those diagnosed with OBPP. The patients significantly differed in all studied parameters.

The assessment of cognitive skills showed that, in the tests for visual memory and thinking, the children diagnosed with AMC and those diagnosed with OBPP differed from the healthy children. Our data support the assumption that impairments in motor functions are associated with impairments in individual cognitive functions. This is consistent with data showing that the development of motor skills correlates with the development of cognitive skills [21,22]. However, the sample size did not allow us to statistically estimate the age at which this correlation is the most prominent. There are age periods in which the active formation of voluntary behaviour regulation, reflection, and self-control occurs [23,24,25]. Moreover, voluntary movement skills are mainly formed in childhood. However, it can be argued that, in general, the formation of these skills is sequential; for example, operating with hands is associated with tasks that a child needs to perform at a particular stage of development, such as bringing food to their mouth, handling various objects, and writing; all these movements are associated with cognitive control of voluntary movements. In addition, it is critical for the formation of consistency in all body functions [22]. Thus, it is important to take this factor into account when developing educational programmes for children with motor disorders.

Previous studies have found a link between memory and motor skills [26,27]. A deviation in the capacity of visual memory and auditory memory was also found in children diagnosed with AMC and those diagnosed with OBPP [16].

Taking this into consideration, we suggest that the difference in the development of cognitive skills may be associated with a basic mismatch in the development of motor and cognitive skills. Based on the literature, correlations were found in the cognitive level of children with impaired formation of hand preference [28], posture control [29], and walking [30], among other impairments [1]. Therefore, on the one hand, it can be hypothesised that any motor impairment must directly lead to cognitive impairment. However, our data are partially inconsistent with this claim since they showed selectivity of cognitive impairment in motor diseases and did not show any differences in memory performance between AMC patients and OBPP, patients despite their prominent differences in the motor domain. In this situation, several factors can be identified that affect the impairment of cognitive skills. Cognitive skills are formed in relation to motor skills, and this relationship could be nonlinear. There may be a “ceiling” effect, when, in the case of restraining a specific motor function, the function of the world perception would be discretely violated and, accordingly, cognitive impairments have a similar severity [31,32,33]. Another important factor, in our opinion, would be the social environment, which is also important for the formation of the patient’s cognitive skills. Although, there may be other reasons for the effects we discovered.

Among the various factors that may affect the complex link between the motor and cognitive domains of development, parenting appears to be of particular importance. The parents of children with motor disorders perform most basic activities, such as dressing, feeding, and washing, for their children from birth. Notably, as the child grows, no psychological separation from the parent occurs; they continue employing the same, albeit no longer age-matching, parenting approaches. It is difficult for parents to step back from such a role. They continue to perform actions for the child in the usual routine, thus inhibiting the child’s physical development. Such a manifestation of overprotection in parents who have a child with a serious illness is common [34]. Several studies have shown that such parental behaviour inhibits both the cognitive and mental development of a child [35,36]. Furthermore, parental overprotection increases the level of anxiety in children, which also suppresses the development of cognitive functions [37,38]. Notably, in most countries, the treatment and rehabilitation of AMC patients and OBPP patients are not associated with the severity of disease manifestation. This may also result in the levelling of the factor of individuality.

The involvement of modern neurotechnology may facilitate an understanding of the psychophysiological changes that occur in the brain in children with motor disorders [39]. It has been shown that, in such children, there is a significant decrease in the power of the main EEG rhythms [40]; moreover, specific rearrangements in the brain that are associated with a change in the functional representation of certain muscles of the upper extremities have been found in children with AMC [41]. It is likely that similar changes should occur in children diagnosed with OBPP, but there is no exact data on this issue. Understanding the relationship between such significant reorganisations in the brain and the level of development of cognitive skills may make it possible to reconstruct the mechanisms of compensatory brain activity in children with motor disorders. This will enable the creation of neurotechnology that can help children with motor disorders. Regular clinical and neurophysiological estimation, an assessment of the needs in daily life, and knowledge of the social and family environments are key points for management.

Accordingly, to eliminate cognitive impairment in children with motor diseases, it is necessary to focus not only on the correction of motor dysfunctions but also on the minimisation of the effects of the children’s social environment.

## 5. Conclusions

This study has yielded the following conclusions:Children with AMC and those with OBPP differ from healthy children in terms of cognitive skills, such as visual memory and thinking.Children with AMC and those with OBPP differ in terms of motor skills but not in cognitive tests.Presumably, the presence of a motor disease may be a more significant factor than its degree of manifestation in explaining cognitive deficits.

## Figures and Tables

**Figure 1 ijerph-20-01841-f001:**
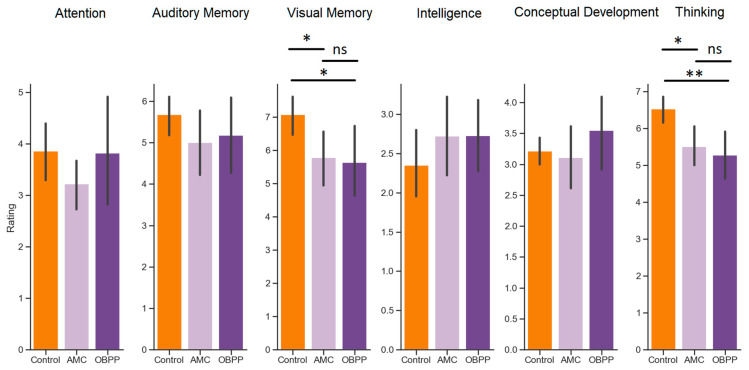
Performance in cognitive tasks assessed for AMC patients (light purple), OBPP patients (dark purple), and control children (orange). Asterisks between the bars indicate the significant difference shown by post hoc Mann–Whitney comparisons between two groups (* *p* < 0.05; ** *p* < 0.01, ns: not significant).

**Figure 2 ijerph-20-01841-f002:**
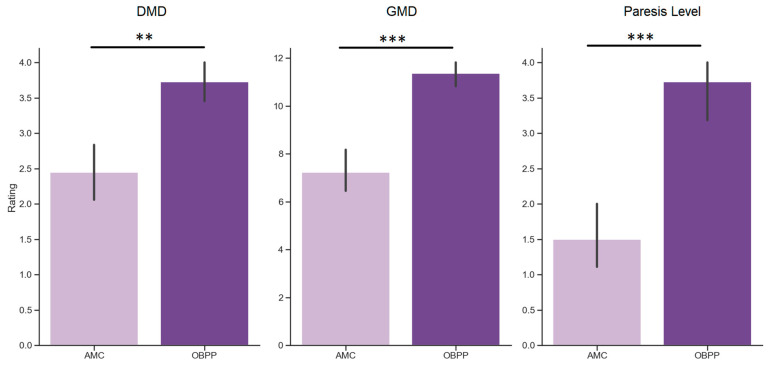
Motor development in AMC (light purple) and OBPP (dark purple) patients, assessed as delayed motor development (DMD) score, general motor development (GMD) score and the level of paresis (Paresis Level). Asterisks indicate significant difference for hoc Mann–Whitney u-test comparisons between the two groups of patients (** *p*< 0.01, *** *p*< 0.001).

## Data Availability

The data presented in this study are available on request from the corresponding author. The data are not publicly available due to patient confidentiality.

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
