# Peer review of "Similar Cognitive Skill Impairment in Children with Upper Limb Motor Disorders Due to Arthrogryposis Multiplex Congenita and Obstetrical Brachial Plexus Palsy"

_ijerph, 2023, doi:10.3390/ijerph20031841_

Round 1

Reviewer 1 Report

Dear authors,

I have read your manuscript 'Similar Cognitive Skill Impairment in Children With Upper 2 Limb Motor Disorders Due to Arthrogryposis Multiplex Congenita and Obstetrical Brachial Plexus Palsy" and have the following comments and suggestions:

- first, the number of included subjects is indeed low and possibly preclude reaching definite conclusions regarding the link between motor and cognitive impairment

- I find the presentation of the results very brief compared to the long introductory part. in addition, from the presented graphs it appears that OBPP and AMC children might be even more intelligent and exhibit improved conceptual development compared to healthy children (although not statistically significant)?

- in line 189, while discussing the link between motor and cognitive impairment you conclude that there is no need to consider social factors, yet you did not assess these factors in the study

- similarly in lines 190-203, in the study you did not assess the parenting style or quantify the overprotective behaviour of parents

- regarding the aspects discussed in line 209-210 - rearrangements in the neural networks are very likely to occur in OBPP patients as well, given the remarkable neuroplasticity of young brains

In addition, some minor spelling errors and unclear word choices:

- in line 37 "clearer" - replaced 

- it is not clear what you meant in line 146 by "higher levels of paresis" - does it refer to the degree of motor deficit or the more proximal situation of affected muscles?

Author Response

Dear authors,

I have read your manuscript 'Similar Cognitive Skill Impairment in Children With Upper 2 Limb Motor Disorders Due to Arthrogryposis Multiplex Congenita and Obstetrical Brachial Plexus Palsy" and have the following comments and suggestions:

- first, the number of included subjects is indeed low and possibly preclude reaching definite conclusions regarding the link between motor and cognitive impairment

We thank Reviewer 1 for this suggestion. In order to assess the sufficiency of the sample size used we have approximated the eta-squared values for the Kruskall-Wallis series with H-values following Murphy & Myors (2014, Appendix A). We have included them into the extended version of Table 1:

Task

h

η2approximated

p

Attention

2.38

0.04

1.000

Auditory Memory

1.83

0.03

1.000

Visual Memory

9.90

0.16

.042 *

Intelligence

1.62

0.03

1.000

Conceptual Development

2.65

0.05

1.000

Thinking

12.76

0.19

.010 *

The sample size, which would be required to demonstrate the effect with a power of 0.8 (α = 0.05), is estimated to include 18 participants per group for Visual Memory performance, and 12 participants per group for Thinking performance.  We had 18 amyoplasia participants and 11 OBPP patients only available for statistical comparison.  Yet, we observed statistically significant effects. The sample size problem would be more difficult  in case we would see no significant difference.  In that case, a minimally sufficient experimental power would make it challenging to determine if the lack of effect is real. We have expanded the table in the results section.

- I find the presentation of the results very brief compared to the long introductory part. in addition, from the presented graphs it appears that OBPP and AMC children might be even more intelligent and exhibit improved conceptual development compared to healthy children (although not statistically significant)?

- We thank Reviewer 1 for this observation. . We added the following paragraphs  to complement the corresponding part of the manuscript with the discussion of the aforementioned observation.

The patients  were examined during their spared time. They attended the laboratory being either supervised by their parents or medical personnel. Their self-care capabilities are significantly different from healthy children. The patients could not take food or perform hygiene actions (wash face and hands) on their own. They rather did so under the patronage of their parents or medical personnel.  We also noticed that all the movements performed by the patients appear to be slower compared to healthy children. We plan to quantify these  skills. Visual communication with healthy children and patients did not differ.

In addition, by more observation, one would not tell the difference in communication between children diagnosed with AMC and OBPP compared to healthy children in communication. That is why we were surprised by the results of psychological testing. This observation became the source of inspiration for us to examine differences and similarities in cognition across  AMC,  OBPP and healthy  children.. Since these statistics are made taking into account the correction for multiple comparisons, the conclusions can be quite critical for understanding changes in cognitive functions in children with motor disorders.

- in line 189, while discussing the link between motor and cognitive impairment you conclude that there is no need to consider social factors, yet you did not assess these factors in the study

- similarly in lines 190-203, in the study you did not assess the parenting style or quantify the overprotective behaviour of parents

Thank you very much for this comment. In this paragraph, we wanted to point out that there is data that directly links motor impairment with cognitive impairment. Of course, the picture of what is happening is much more complicated. On the one hand, There are individual differences in the neural networks subserving neuroplasticity of cognition function mediated by motor impairment. On the other hand, there is a social factor that compensates for the body's incapabilities. To our best knowledge, there is literally no studies available where such interplay would be studied or these two factors were clearly separated. That is why we give quite a lot of space to the social factor in the discussion of the results. The similarity of cognitive impairments also indicates a large role of the social factor. We have changed “Hence, it seems impossible to directly link motor disorders and cognitive impairments without considering various social factors.” to:

There may be a “ceiling” effect, when, in the event of a violation of a specific motor function, the functions of world perception are simply discretely violated and, accordingly, cognitive impairments have a similar severity [31–33]. Another important factor, in our opinion, is perhaps the social environment, which is also important for the formation of the patient's cognitive skills. Although, there may be other reasons for the effects we discovered.

- regarding the aspects discussed in line 209-210 - rearrangements in the neural networks are very likely to occur in OBPP patients as well, given the remarkable neuroplasticity of young brains

Thank you for your comment. Of course, there are such rearrangements in patients with OBPP. However, we did not find any data in support of their proposition. Nevertheless, following your suggestion, we now have included in the article the following:

It is likely that similar changes should occur in children diagnosed with OBPP, but there is not much data supporting this proposition. In the light of this hypothesis, further studies of the cognitive development in OBPP would be necessary to perform.

In addition, some minor spelling errors and unclear word choices:

- it is not clear what you meant in line 146 by "higher levels of paresis" - does it refer to the degree of motor deficit or the more proximal situation of affected muscles?

We thank the Reviewer for this clarification question.   A lower level of paresis is associated with greater motor impairment in the patient, i.e. greater involvement of the distal muscles.

We estimated passive and active movement in the joins, muscles strength, muscles volume, muscles tone, tendon reflexes, and sensation. The paresis level was estimated clinically by the scheme of segmental innervation of upper limb muscles.

 We changed the paragraph accordingly:

During neurology examinations, the patient’s general motor skills (GMD) were estimated. Anamnesis vitae included information about the patient in infancy. A developmental delay was indicated when a child had not reached particular milestones within the expected time period and his motor skills were differ from a healthy child (DMD). We estimated passive and active movement in the joins, muscles strength, muscles volume, muscles tone, tendon reflexes, and sensation. The paresis level was estimated clinically by the scheme of segmental innervation of upper limb muscles. A lower level of paresis is associated with greater motor impairment in the patient, i.e. greater involvement of the distal muscles.

Reviewer 2 Report

I believe the paper address very interesting point on the relation of motor disfunction and cognitive skills and represnt sound research on a group of children with AMP and OBPP. At the same time a have few concerns.

*I belive the novelty od the study should be more emphasized as reference 16 also showed visual memory disfunction in AMP. Weather it is adding anorther group of patients or other test, etc it should be stated more explicitly.

*Authors spend quite a lot of time in the discussion for social factors contribution into cognitive skills development in children with disabilities although these research does not examine these factors at all. I believe that, first of all, not only social factors can explain the observed results and secondly, if authors belive that their contribution is most important they should have beed check it somehow. Otherwise I suggest to shorten this discussion and just point it as one of the potential fuctors that should be examined in the future research/

*I believe term "sick children or kids" is not appropriete here as it generally refer more to some acute condition, but AMP an OBPP is more long-term.

Author Response

I believe the paper address very interesting point on the relation of motor disfunction and cognitive skills and represnt sound research on a group of children with AMP and OBPP. At the same time a have few concerns.

*I belive the novelty of the study should be more emphasized as reference 16 also showed visual memory disfunction in AMP. Weather it is adding another group of patients or other test, etc it should be stated more explicitly.

-Thank you for this note. Indeed, in one of our previous studies, we found that children with AMC differ from healthy children in cognitive performance. In this study, we included a group with a similar diagnosis of OBPP (in terms of symptoms and treatment). This was necessary to study the question of how the severity of the disease correlates with the severity of cognitive impairments. In our opinion, this is a completely different study and is a continuation of the previous one. To emphasize this more explicitly, we added to the introduction:

We also correlated the severity of the motor impairment with the level of cognitive performance.

*Authors spend quite a lot of time in the discussion for social factors contribution into cognitive skills development in children with disabilities although these research does not examine these factors at all. I believe that, first of all, not only social factors can explain the observed results and secondly, if authors belive that their contribution is most important they should have beed check it somehow. Otherwise I suggest to shorten this discussion and just point it as one of the potential fuctors that should be examined in the future research/

-Thank you very much for this comment. In this paragraph, we wanted to point out that there is data that directly links motor impairment with cognitive impairment. Of course, the picture of what is happening is much more complicated. For the first,  There might be huge  individual  variability of neuroplasticity  in the neural networks subserving motor and cognitive functions we test. On the other hand, there are a number of social factors that compensate for the body's incapabilities. We practically do not find studies where such an influence would be clearly separated. That is why we give quite a lot of space to the social factor in the discussion of the results. The similarity of cognitive impairments also indicates a large role of the social factor. We have changed “Hence, it seems impossible to directly link motor disorders and cognitive impairments without considering various social factors.” to:

There may be a “ceiling” effect, when, in the case of restraining a specific motor function, the function of the world perception would discretely violated and, accordingly, cognitive impairments have a similar severity [31–33]. Another important factor, in our opinion, would be  the social environment, which is also important for the formation of the patient's cognitive skills. Although, there may be other reasons for the effects we discovered.

and

It is likely that similar changes should occur in children diagnosed with OBPP, but there is no exact data on this issue.

We will also be very grateful to the reviewer for additional versions explaining the results obtained.

*I believe term "sick children or kids" is not appropriete here as it generally refer more to some acute condition, but AMP an OBPP is more long-term.

Thank You. We fixed it.
